# Effects and Side Effects in a Short Work Coaching for Participants with and without Mental Illness

**DOI:** 10.3390/bs14060462

**Published:** 2024-05-30

**Authors:** Lilly Paulin Werk, Beate Muschalla

**Affiliations:** Department of Psychotherapy & Diagnostics, Institute of Psychology, Technische Universität Braunschweig, Humboldtstraße 33, 38106 Braunschweig, Germany; b.muschalla@tu-braunschweig.de

**Keywords:** workplace intervention, individual coaching, unwanted events, work ability, work capacities, coping behavior, mental disorders

## Abstract

Employees with mental illness are often the first to be unable to cope with increasingly complex psychosocial work demands. But people without mental illness can also suffer from, for example, high workload. This study compares a short coaching to stabilize work ability for employees with and without mental illness regarding coaching topics, effects on work-related resources, goal attainment, and unwanted events. Individual coaching of three sessions (problem exploration by behavior analysis, practice of new behavior, reflection) was conducted with employees from different professional fields. A medical history was taken to determine whether participants are affected by a mental disorder. All coaching was conducted by the same behavior therapist in training (L.P.W.) under the supervision of an experienced behavior therapist (B.M.). Two hundred and three coachings with three sessions were completed. In total, 103 participants did not have a mental illness (51%), and 100 participants reported a mental disorder (49%). The coaching participants with mental illness had lower initial levels of work-related capacities (more severe impairments) and coping behavior as compared to the participants without mental illness. In the pre–post comparisons, both groups achieved significant improvements in work-related coping after the coaching. There were no differences in goal attainment between both groups. While participants without mental illness reported more unwanted events in parallel to the coaching (30% reported negative developments in life), participants with mental illness reported coaching-related unwanted events (20% felt to be dependent on the coach). Coaching with an individual focus on one topic can improve work-related resources in participants with and without mental disorders. Since participants with and without mental illness experience different unwanted events in coaching, psychotherapeutic expertise is needed in order to set the right focus.

## 1. Introduction

Work is one of the most important areas of life for experiencing self-efficacy [1] and life satisfaction [2]. For good job satisfaction and work ability, an employee’s capacities must match the demands of their job (person–job fit [3,4,5,6,7,8,9,10]). Work ability can be defined as “the balance between human resources and the demands of work” [11] but should always be related to a reference context (e.g., the current workplace or the general labor market) [12]. In times of high social and cognitive work demands, work ability becomes increasingly challenged. Employees with mental disorders are often the first who are no longer able to meet these increasing work demands, so work ability problems and absenteeism increase [13,14,15,16,17,18,19,20]. But healthy employees also report increased workload due to digitalization, computer-monitoring, and benchmarking [21,22,23,24,25].

In order to keep employees’ work ability and job satisfaction, it is important to create a good individual person–job fit [3]. Workplace adjustments for adapting work demands to the employee are not possible in all work areas [26]. Individual problem detection and solving by coaching employees’ psychosocial capacities is an alternative [27]. An effective intervention for stabilizing work ability is coaching individual capacities [28,29,30,31,32]. Work-related coaching may be helpful for the person’s life in general.

Many studies [33,34,35,36] evaluate coaching in work settings without consideration of mental health status. Until now, it was unclear whether there are differences in the coaching of participants with and without mental illness. From a clinical point of view, this is, however, important, since persons with mental illness have other problems and need different interventions as compared to employees without mental illness [37,38,39].

Against this background and lack of research, this present study investigates short individual work coaching in participants with and without mental illness. Instead of well-being, job satisfaction, or symptoms, the focus here is on behavioral resources (work ability, work-related capacities, work-coping behavior), which are necessary to cope with individual work problems. Goal attainment and unwanted events were monitored as well.

### 1.1. Effects of Individual Work Coaching

It is known that coaching is an effective intervention: the effects of individual coaching have been reported in meta-analyses [40,41] with a significant positive impact on affect, work attitudes, self-regulation, skills, and work performance. After coaching, observed work performance has been reported to be higher than before coaching [42]. There were no significant differences in effectiveness between different application types (face-to-face, videophone [40]). Comparing externally commissioned coaches outside the organization and organization-internal coaches, slightly weaker effects emerged for external coaches [40]. The frequency and number of coaching sessions had no significant effect on the overall effects of coaching [40]. Coaching addresses different target groups. Business coaching on productivity and work performance for senior workers also had partly positive effects on self-efficacy, resilience, and well-being [43]. Systematic reviews [43,44] also ask for coaching mechanisms, but the studies in this field are quite heterogeneous. The consensus for successful coaching seems to be a mixture of appropriate methods and techniques (“teaching practice”) and relationship quality between the coach and coachee. For the coach, facilitative behavior, similarity of coach and coachee, and strength-based methodology could play a role in coaching success [43]. The quality of the coach is mentioned in many studies but operationalized differently. For example, definitions range from the promotion of psychosocial capacities [45,46] to communication strategies [47] to specific coaching skills [46,48,49]. Coachee characteristics such as engagement [44], self-efficacy, and learning styles are thought to influence the coaching process [43]. Forming a kind of “working alliance” seems to increase the effects of coaching, as well as rapport, trust, and commitment [43]. A systematic review emphasizes the importance of the coaching relationship and the influence of coaches’ attributes on the intervention’s effects [50].

### 1.2. Participants with and without Mental Illness in Coaching

Work coaching for employees without mental illness often targets topics like the self-structuring of work (e.g., dealing with procrastination [51]), role clarification (e.g., leadership behavior [52]), improving social skills (e.g., dealing with colleagues [53,54,55]), and career topics (e.g., career enjoyment [56]). The topics are concretely named by the participants and adapted for the coaching together with the coach [57].

People with mental illness also come to coaching due to the low-threshold access and the shorter waiting times compared to psychotherapy [58,59]. To an untrained eye, mental disorders are not always visible at first glance, and the coaching concern is usually not “dealing with my mental disorder”, so mental illness remains unnoticed in coaching. In the general population, the difference between psychotherapy (health treatment under strict quality and training standards [60]) and coaching (non-protected term without quality criteria [58]) is often not clear: “Coaching is not a healing profession. In current definitions of coaching, in contrast to psychotherapy, the treatment of mental disorders is explicitly excluded” [61].

Coaching can also fit in with the complaints of people with mental illness, as the level of functioning in multiple areas of life can be reduced due to disorder-related capacity impairments [62,63]: In family life, a person with mental illness may experience problems in their partnership due to interaction problems or social withdrawal, so the person signs up for life coaching to save their partnership. At work, people with mental disorders may experience conflicts with colleagues or may be overwhelmed by the workload and register for work coaching. In two studies, Grant’s working group investigated how many life-coaching participants met a clinically relevant number and severity of symptoms of mental stress in the Hopkins Brief Symptom Inventory and found 52% [64] and 26% [65]. So, we know that people with mental illness come to coaching, but we know little about how coaches deal with it [66,67].

The first step in the beginning of the coaching should be to determine whether there are mental disorders that could influence the coaching. Coaches are advised to acquire basic knowledge about mental disorders, their symptoms, and treatment [59,68]. It is recommended that coaches seek advice from psychotherapeutic specialists and make use of supervision during the diagnostical and practical process [58,59,69]. Coaching expertise also includes recommending suitable treatment providers for mental disorders and supporting participants in making contact [58,59]. If mental disorders are overlooked or coaching acts contrary to ongoing psychotherapy, coaching can become harmful by triggering or exacerbating problems in the first place [58,70]. This would be the case if overly ambitious coaching goals led to increased self-deprecation and hopelessness in a depressed participant [71], or if relaxation is practiced in coaching to reduce anxiety during an ongoing exposure treatment in psychotherapy [72]. To prevent negative coaching effects (unwanted events), Schermuly (2016) recommends close coordination between coaches and therapists (with participant’s consent) when psychotherapy is running in parallel [70]. It is advisable to set a sufficiently different coaching goal than psychotherapy to differentiate the interventions from one another [59,73].

In a previous publication examining this sample here [72], the authors investigated whether the coaching of participants with and without mental illness differed qualitatively: while participants with mental illness required support in care coordination or workplace adaption beyond symptom management (e.g., establishing contact with treatment providers), participants without mental illness were able to work on a specific work-related topic (e.g., work reduction through self-structuring using a weekly schedule [72]). Further research is needed on the differences between participants with and without mental illness in coaching.

### 1.3. Unwanted Events and Side Effects in Coaching

Health interventions must also consider negative intervention effects, as it can be assumed that everything that has effects also produces negative effects [74]. “Unwanted events” have so far mainly been studied for psychotherapy and are defined as “all events that occur parallel to treatment in relation to the patient” (in the sense of Linden [75]). They occur in at least 3–15% [76] resp. 5–20% [77] of psychotherapy cases. Unwanted events are an over-categorization of side effects. By definition, side effects presuppose that they are directly related to the intervention, which is often difficult to trace [75,77]. If there are conflicts with a colleague at work and the worries about the colleague increase during coaching, it is not so easy to distinguish whether the worrying is worsening because of the ongoing collegial conflict or because of the coaching.

Unwanted events and side effects in coaching are little researched [78]. Schermuly et al. [79] have provided an initial definition, influencing factors, and exploratory data on types of negative coaching effects [79]. They define coaching side effects as “harmful or undesirable consequences for the client that are caused by correctly conducted coaching, and that occur in parallel or subsequently” (p. 19). The coach–coachee relationship, time and place, communication, and motivation aspects may come along with negative effects during or after coaching [34,79]. Addressing side effects can help coaches to deal with challenging situations and act in accordance with ethical standards [80]. If potential side effects of interventions are openly addressed with participants, they can be reduced or even prevented and do not negatively impact the alliance [79,81].

A systematic review [82] identified side effect types for participants, coaches, and organizations. For participants, there is a danger that mental health problems cannot be adequately addressed during coaching and that the power imbalance is exploited by the coach [71]. Long-term negative consequences for participants could be job loss, financial losses, psychological problems, low motivation, and well-being [78]. For coaches, it is difficult not to be able to influence the long-term impact of the coaching, if coaching topics affect them personally or if they cannot meet their expectations in the role of a coach [82,83]. Coaching also appears to have side effects at the organizational level [82,84]. Participants may develop in other directions than desired by the organization and thus fail to meet organizational conditions. This can result in conflicts with supervisors and colleagues, and a possible loss of reputation [82,84].

The severity and duration of side effects can vary greatly. Short-term side effects concern a temporary drop in job satisfaction because complaints that become conscious or changed behavior and thinking at work consume more resources or lead to conflicts. Long-term effects concern a dependency or more difficult detachment from the coach [85].

Participants report side effects quite frequently but of low to moderate intensity [86]. In contrast, little is known about the causes and stability of side effects. Some studies found that relationship problems or (missing) supervision may support the occurrence of side effects. It is also possible that short-term negative side effects can turn into something positive in the long term, as, for example, conflicts are resolved after they have been addressed and are viewed positively in retrospect [85].

## 2. Materials and Methods

### 2.1. Research Questions

As can be seen from the above reviewed literature, there are already numerous RCT studies and systematic reviews on individual coaching, but these often focus on state variables like well-being, mood, or psychological complaints [40,41,42]. In this study, the approach of behavioral outcome variables of work ability and coping strategies was chosen to measure coaching effects. It examines what positive effects on work-related resources three sessions of coaching can achieve in participants with and without mental illness. Coaching goal attainment and unwanted event occurrence are compared for participants with and without mental illness.

The following research questions were investigated:Which topics are dealt with in coaching with participants with mental illness as compared to participants without mental illness?How do work ability, impairments in work-relevant capacities, and work-related coping behavior change pre–post three coaching sessions and two weeks after in participants without mental illness as compared to participants with mental illness?How do the coach and the participants (with and without mental illness) rate the degree of coaching goal attainment?Which unwanted events occur in coaching participants with and without mental illness, and how often do they occur?

### 2.2. Procedure

From 2021 to 2023, 203 coaching interventions (each with 3 sessions) were conducted as part of the EU project H-WORK [87]. Among 223 registrations, there were 11 non-participants and 9 drop-outs. The coaching consisted of three sessions of one hour each after voluntary registration. Participants were recruited via announcements in organizations, newsletters, and boards (mainly employees of a public university and a public hospital) and a public call in a newspaper. All coaching was conducted by the same psychologist in training as a behavior therapist (L.P.W.) under the supervision of an experienced behavior therapist (B.M.) in an online or face-to-face-setting. One specific topic was chosen for each coaching, derived from an initial situational behavior analysis using the SORKC model [88]. Between the coaching sessions, there were usually 2 to 4 weeks to conduct exercises in the workplace.

### 2.3. Work-Related Coaching

The coaching was conducted according to the behavior-oriented training manual “Chose one—act on one! A Three Session Coaching on a Selected Work Problem” [89].

The first session started with a diagnostic exploration of professional and medical history. Based on the standardized screening question “Are you currently suffering from health problems that are not purely physical but also of a mental nature (e.g., enduring and impairing depressive mood, or anxiety, or difficulties in everyday interaction with others)?” and the exploration of (work) participation problems and previous treatments, the participants were divided into the groups “participants without mental illness” and “participants with mental illness” participants. Then, the work-related problem was explored using behavior analysis. An individual coaching goal was set together with the participant.

The second coaching session involved practicing new cognitive and/or behavioral behaviors, using individually selected coaching techniques [90]. In most cases, a homework task was given in which the participants should practice the newly learned behavior or cognitions in the workplace.

The third coaching session served for planning practical implementation in everyday work (what are facilitators and how can I cope with barriers?) and to reflect on the coaching. One example would be a participant who works overtime every day due to structuring problems and is therefore burdened. The coaching goal could be to finish work on time once a week. Coaching would use concrete plans to improve the self-organization of a working day.

### 2.4. Evaluation

Participants completed questionnaires in the beginning of the first coaching session (T0), at the end of the third coaching session (T1), and two weeks after the third coaching session (T2, see Table 1). For economic reasons, the capacity impairments (Mini-ICF-APP-S) were only recorded pre (T0) and post (T1). T0 and T1 were completed in paper–pencil versions during coaching. T2 was a retrospective online survey with a low response rate (*N* = 30), so calculations were carried out for T0–T1 and T0–T2. In co-variance analyses (ANCOVA) and multivariate analyses of variance (MANOVA), the time course was examined for significant changes in the work-relevant resources, considering the control variables age, gender, and mental health status.

#### 2.4.1. Work Ability Index (WAI [91])

The Work Ability Index (WAI [91,92]) was used in the validated German version to assess the subjective work ability of respondents in relation to their current, last, or similar job and the demands associated with it. In this study, the following item was used to self-assess current work ability: “Current work ability compared with the lifetime best: Assume that your work ability at its best has a value of 10 points. How many points would you give your current work ability? (0 means that you cannot currently work at all)”. The scale ranges from 0 = completely unable to work to 10 = best work ability. The WAI was validated using a representative sample of employed persons in Germany (Cronbach’s α = 0.75 [93]). In this sample, only one item was used, which has been evaluated successfully before [94].

#### 2.4.2. Mini-ICF-APP-S [95]

To explore employees’ self-assessment of impairments in work-related capacities, the original German version of the Mini-ICF-APP-S was used. Thirteen work-related capacity dimensions were rated on a scale from 0 = this is clearly a strength of mine, to 3 = this is somehow possible and 4 = this does not always work, to 7 = I am fully unfit. The score can thus be interpreted as an impairment scale. The Mini-ICF-APP is based on established psychological capacity concepts. It is an internationally validated [96,97,98] and widely used instrument for work ability assessment and is recommended in several social medicine guidelines [99,100,101] for objectifying work ability status. The Mini-ICF-APP-S has been validated in different patient samples and shows a good inter-rater reliability (r = 0.92 [27]) and validity in the sense of correlations with the work ability and a sensitivity to change [102]. For the Mini-ICF-APP, good internal consistencies of Cronbach’s α = 0.92 were reported for the sample of patients with different psychiatric diagnoses [103]. In the sample investigated here, the reliability of the 13 Mini-ICF-APP-S items was Cronbach’s α = 0.76. The 13 capacity dimensions are adherence to regulations, planning and structuring of tasks, flexibility and ability to adapt to changes, competency and application of knowledge, ability to make decisions and judgments, proactivity and spontaneous activity, endurance and perseverance, assertiveness, contact with others and small talk, group integration, dyadic or close relations, self-care, and mobility [95].

In this study, a mean score was taken over all 13 capacity dimensions. This is an indicator of the overall level of work-relevant psychological capacities. The Mini-ICF-APP-S was completed at T0 and T1. An ANCOVA with the control variables gender, age, and mental health status was calculated for the comparison between both measurement points.

#### 2.4.3. Inventory for Job Coping [104]

The Short Inventory for Job Coping (JoCoRi [104]) is an originally German measurement tool and asks for active work-related coping behavior in an economic and differentiated way. The abilities for work-related self-calming and self-instruction (four items) as well as active problem solving and interaction at work (three items) are rated on a Likert scale from 0 = not able to do this to 4 = best coping for doing this. Respondents are mentally placed in their workplace by the instruction “Please imagine being at your workplace right now. How could you do the following things?”. A mean score is calculated over all 7 items; it can be between 0 and 4. The JoCoRi was validated in heterogenous mid-aged employees from diverse professional fields who also had different orthopedic, cardiology, neurology, and psychosomatic complaints. Satisfactory internal consistencies ranging from α = 0.83 to 0.91 were obtained for these samples [104]. In this present study, the internal consistency of the JoCoRi was Cronbach’s α = 0.68.

#### 2.4.4. Coaching Goal Attainment [105]

After a coaching goal was defined together with the coach at the end of the first coaching session (based on the SMART principle: Specific, Measurable, Attractive, Realistic, Terminated [106]), it was evaluated separately by the participant and the coach at the end of the third session using a goal attainment scale (according to Turner-Stokes [105]). The predefined coaching goal was assessed on a five-point scale: 1 = much less than expected, 2 = less than expected, 3 = expected result, 4 = more than expected, and 5 = much more than expected. Goal-attainment scaling is frequently used in rehabilitation therapy, where it has shown moderate (κ = 0.48 [107]) to almost perfect (κ = 0.97 [108]; κ = 0.63 [109]) inter-rater reliabilities in different clinical samples [110].

#### 2.4.5. Unwanted Events/Adverse Treatment Reactions Scale [75]

The Unwanted Events/Adverse Treatment Reactions Scale (UE-ATR-PT [111]) is a self-assessment questionnaire for the assessment of unwanted events in individual behavior-oriented interventions. Unwanted events perceived by the coachee are assessed, i.e., events during coaching that lead to negative effects. This also includes side effects triggered by the coaching (e.g., stress in coaching sessions) as well as negative effects that occur in parallel (e.g., exhaustion due to a cold). In this present study, the participant perspective was captured in 14 items. The wording in some items was adapted to coaching (i.e., “therapist” was replaced by “coach”, “therapy” was replaced by “coaching”). The following unwanted events were assessed: (1) worsening of existing complaints and symptoms, (2) occurrence of new complaints and symptoms, (3) aggravation of problems, (4) dependence on the coach, (5) discomfort during coaching sessions, (6) problems with the coach, (7) dissatisfaction with coaching result so far, (8) more sessions needed, (9) problems in partnership and family, (10) problems with friends, (11) neighbors or other people, (12) problems at work, (13) other negative developments in life, and (14) problems with third-party coaching comments. A distinction was made between 0 = did not occur and 1 = occurred. The questionnaire was completed at the end of the third coaching session (T1) by the participants. In this sample, the internal consistency of the side effects scale was Cronbach’s α = 0.67. This indicates that items do not all correlate highly, which is plausible, as the items have different contents.

### 2.5. Sample Description

The coaching was announced in regional media and to regional companies (production and technology, education, administration, clinical and health) within a middle city (250,000 inhabitants) in Germany. Participants applied by themselves. Coaching participants thus came from a wide range of professional fields (see Table 2). There were no significant differences in the frequency distribution of professional fields between participants with and without mental illness: Around a third of both groups work in office jobs, another third in service jobs. Healthy participants were slightly more likely to come from education and research (without mental illness: 25%, with mental illness: 20%), while participants with mental illness were more likely to come from the healthcare sector (without mental illness: 9%, with mental illness: 18%). Participants from production were poorly represented in both groups, with 6 and 4 people.

## 3. Results

### 3.1. Coaching Topics in Employees with and without Mental Illness

The distribution of coaching topics is similar in both groups: Workload (30% in both groups) and interpersonal demands (without mental illness: 36%, with mental illness: 37%) make up the largest topics, followed by career issues (without mental illness: 12.5%, with mental illness: 10%). Role stressors and lack of control are slightly more common among healthy participants (without mental illness: 6.5%, with mental illness: 4%), whereas problems with job conditions are more common among participants with mental illness (without mental illness: 3, with mental illness: 9%). Situational constraints only occurred in 4% of both groups. Acute stressors did not occur in either group (see Table 2).

### 3.2. Change in Work-Related Resources Pre–Post Coaching and Two Weeks after the Third Coaching Session

Work ability was rated as significantly higher after than before coaching: Multivariate Analysis of Variance (MANOVA) shows significant mean differences between T0 and T1 with a small effect size (M_T0_ = 6.50, M_T1_ = 7.22; *F*(202) = 8.23, *p* = 0.005, η^2^ = 0.040, Figure 1). In the MANOVA for participants with full follow-up data (T0-T2), the effect is no longer visible (M_T2_ = 7.90, *F*(29) = 0.60, *p* = 0.551, η^2^ = 0.023, Figure 1). Pairwise comparisons show a significant difference in work ability between T0 and T2 (M_Diff_ = 1.03, 95%-CI [0.27, 1.80], *p* = 0.005, *N* = 30) but not between T1 and T2 (M_Diff_ = 0.07, 95%-CI [−0.64, 0.78], *p* = 1.00, *N* = 30). Gender (*F*(1,202) = 0.14, *p* = 0.709, η^2^ = 0.001), age (*F*(1,202) = 2.48, *p* = 0.117, η^2^ = 0.012), and mental health status (*F*(1,202) = 2.06, *p* = 0.152, η^2^ = 0.010) show no interaction with work ability over time (Figure 1).

After coaching, the capacity impairments were rated as significantly lower than before coaching: the ANCOVA shows significant mean differences between T0 and T1 with a large effect size (M_T0_ = 2.00, M_T1_ = 1.66, *F*(1,202) = 32.95, *p* ≤ 0.001, *N* = 203, η^2^ = 0.144, Figure 2). Mental health status initially differed significantly between the participants without and with mental illness (*F*(1,202) = 13.40, *p* ≤ 0.001, *N* = 203, η^2^ = 0.064): At the beginning of the coaching (T0), participants with mental illness showed significantly stronger capacity impairments (*F*(1,202) = 9.13, *p* = 0.003). After coaching (T1), the groups no longer differed significantly from one another (*F*(1,202) = 1.59, *p* = 0.209, Figure 2).

For work-related coping behavior, variance analysis with repeated measurement (without control variables) shows significant mean differences in the pre–post comparison (T0-T1) with a large effect size (M_T0_ = 2.82, M_T1_ = 3.08, *F*(1,202) = 60.20, *p* < 0.001, η^2^ = 0.232; Figure 3). Including the control variables gender, age, and mental health status, the ANCOVA yields no significant mean differences between T0 and T1 (M_T 0_= 2.82, M_T1_ = 3.08, *F*(1,202) = 3.02, *p* = 0.084, η^2^ = 0.015; Figure 3). For mental health status, there is the tendency towards an interaction effect (*F*(1,202) = 6.33, *p* = 0.013, η^2^ = 0.031): At the beginning of the coaching (T0), participants with mental illness showed significantly lower coping behavior than participants without mental illness (*F*(1,202) = 7.33, *p* = 0.007). After coaching (T1), the groups no longer differed significantly from one another (*F*(1,202) = 0.38, *p* = 0.541).

Gender (*F*(1,202) = 0.44, *p* = 0.636, η^2^ = 0.014), age (*F*(1,202) = 0.88, *p* = 0.411, η^2^ = 0.028) and mental health status (*F*(1,202) = 1.45, *p* = 0.242, η^2^ = 0.045) show no interaction with coping behavior over time (Figure 3).

### 3.3. Coaching Goal Attainment

Overall, the assessments of participants’ and the coach’s goal attainment do not differ significantly between participants with and without mental illness. Participants without mental illness stated “less than expected” slightly more frequently in their self-assessment than participants with mental illness (without mental illness: 10%, with mental illness: 3%). The remaining options were equally represented in the participants’ goal assessment in both groups, with over 80% of participants in each group stating “expected result” or “more than expected” (see Table 3).

The coach indicated “less than expected” (without mental illness: 2%, with mental illness: 24%) and “expected result” (without mental illness: 13%, with mental illness: 57%) more frequently for participants with mental illness than for participants without mental illness. The coach indicated “more than expected” (without mental illness: 69%, with mental illness: 16%) and “much more than expected” (without mental illness: 16%, with mental illness: 1%) more frequently for participants without mental illness than for participants with mental illness (see Table 3).

### 3.4. Unwanted Events in Coaching

Participants without mental illness were more likely to report no unwanted events: 40% reported no unwanted event, compared to 33% of participants with mental illness. Participants with and without mental illness reported different unwanted effects. Perceived dependence on the coach was reported significantly more frequently by participants with mental illness (20%) than participants without mental illness (4%). There was a tendency towards significant differences in other negative developments in life, which was reported twice as often by participants without mental illness (30%) as compared to participants with mental illness (14%). The worsening of existing complaints (approx. 15% each) and occurrence of new complaints (approx. 16% each) were reported equally often by participants with and without mental illness. The aggravation of problems (without mental illness: 20%, with mental illness: 23%), discomfort during coaching sessions (without mental illness: 7%, with mental illness: 10%), problems with coaching requirements (without mental illness: 6%, with mental illness: 10%), and more sessions needed (without mental illness: 14.5%, with mental illness: 24%) were mentioned slightly more often by participants with mental illness. Problems in partnership and family (without mental illness: 9%, with mental illness: 6%), problems with friends, neighbors, or others (without mental illness: 3%, with mental illness: 2%), problems at work (without mental illness: 25%, with mental illness: 22%), dissatisfaction with coaching result so far (without mental illness: 6%, with mental illness: 1%), and problems with third-party comments (without mental illness: 3%, with mental illness: 1%) were mentioned slightly more frequently by participants without mental illness (see Table 4).

## 4. Discussion

### 4.1. Coaching Topics

In line with the literature, there were no significant differences in the frequency of coaching topics between participants with and without mental illness, as high workload and interpersonal demands are the main stressors in many workplaces [114]. Career issues and role stressors occurred somewhat more frequently among the participants without mental illness. This could be explained by the fact that employees without mental illness are concerned with career development and role clarification, e.g., in management positions, whereas employees with mental illness struggle with the status quo at work (e.g., [115]). Among participants with mental illness, dissatisfaction with job conditions occurred somewhat more frequently, which could be due to changes in work capacities in comparison with the job demands: if capacity impairment due to mental illness cannot be sufficiently solved by the training of the person, then changes in the workplace must be considered as a next possibility for restoring the person–job fit [27,116].

### 4.2. Effects of Coaching

Over the course of coaching, we found improvements in work ability, work-relevant capacities (as impairments could be reduced), and work-related coping behavior for participants with and without mental illness. These results support previous research on the resource-enhancing effects of work-related health interventions: A systematic review on the impact of workplace interventions on work ability covering 17 RCT studies (12 studies with individual coaching) found small effects for a positive impact on work ability [29]. Other studies provide evidence of reduced sick-related absenteeism [117,118]. A newer systematic review of 47 RCT studies showed that individual and counseling services with fewer than 10 sessions were most effective for reductions in absenteeism, which speaks in favor of short interventions, like the coaching presented here [32]. For the choice of setting, individual interventions generally show similarly good effects as group trainings [119]. In an experimental comparison of individual coaching, self-coaching, and group training (same contents, different settings), individual coaching was superior even to group training in terms of goal achievement [120]. Procrastination was improved through individual coaching and group training in equal measure [120]. In practice, a combination of individual and group interventions is recommended [121]. Little is known to date about coaching effects at the level of (work-relevant) capacities, as most studies have relied on well-being and symptoms. It was pointed out that coaching evaluation should increasingly consider capacities (especially work and social functioning) in the sense of the “Theory of Patient Capacity” [122]. This does not only refer to the professional capacities: especially soft skills such as communication and self-management are becoming more and more relevant in many professions [123,124]. Improving capacities in coaching to solve correlated work problems [27] is an important approach to improve the individual person–job fit [3].

Participants with mental illness reported a significantly lower initial level in work-related capacities (more severe impairments were present) and coping behavior, which, however, approached that of healthy participants after the coaching. These results fit in with previous research on the capacity impairments of people with mental illness [125,126], which are particularly relevant in the work context. Coaching appears to be suitable for improving capacity impairments and coping behavior for participants with and without mental illness.

The participants’ rating of goal attainment (over 80% “result as expected” or “more than expected”) indicates that both participants with and without mental illness were able to benefit from the coaching in terms of self-perception. The coach’s goal assessment differs from this in that “less than expected” and “expected result” are more common in the case of participants with mental illness, while “more than expected” and “much more than expected” were stated more frequently for participants without mental illness. It would be interesting for further studies to find out qualitatively (e.g., through interviews) why this is the case. It is possible that there was an expectation in advance that the coaching of participants with and without mental illness would differ less from one another: by focusing on treatment coordination or symptom management in terms of content for many participants with mental illness [72], the work topics (set as goals) were often not focused on, as the underlying mental impairments took priority. In general, but especially in the case of participants with mental illness, the coach should remain flexible in the design of the coaching process and the adaptation of the coaching goal [127,128].

### 4.3. Side Effects

In total, 40% of the participants without mental illness and 33% of the participants with mental illness perceived relevant side effects. There were some differences in the distribution of the type of unwanted events between the participants with and without mental illness. While participants with mental illness were more likely to describe unwanted events in connection with the coaching process (aggravation of problems, discomfort during coaching sessions, problems with coaching requirements), participants without mental illness were more likely to describe external problems with other people and other negative developments in life in parallel to the coaching. This could indicate that participants with mental illness experience more “side effects” in the sense of negative effects triggered by the coaching [79], or that the format of the coaching is not sufficient for people with mental illness (e.g., meeting the requirements, three sessions only).

Comparing the coaching evaluated here with a German routine behavior psychotherapy of 12 to 80 sessions, this coaching was significantly shorter and pursued different goals, which may have meant that participants’ expectations could not be met [69,73]. In particular, the strongest differences in the dependence of participants with mental illness on the coach make clear that participants with mental illness experience coaching differently than participants without mental illness. The coach must take this into account. Three hours of coaching did not seem to be enough for many of the participants with mental illness, which is why the coach should take time to initiate contact with other practitioners (e.g., occupational physician), if necessary, or to derive follow-up interventions like rehabilitation [59,72]. The detoxification process from the coach should also be tackled and initiated at an early stage, as the feeling of dependency can occur after just three sessions [129]. It is surprising that both groups differ quite strongly in the presence of other negative developments in life. It may be that this is actually a heterogeneous factor in both groups or that participants without mental illness attribute complaints more to negative developments that occurred in parallel, independently of coaching. The coaching was mainly carried out during the COVID-19 pandemic, and negative developments in life (e.g., covid infection, closed daycare centers) may have been more present than normally. Work problems were quite common in both groups, which fits in with the fact that the coaching was offered for work-related problems. It is possible that these work problems became even more apparent for the participants through the intensive discussion in the coaching sessions. Coachees thought about their work problems more intensively than they would have done without coaching. Changes in behavior (which were often part of the homework task) may also trigger some new conflicts with supervisors and colleagues, so new work problems may be perceived [82,84].

### 4.4. Strength and Limitations

In this study, a naturalistic convenience sample was investigated. As the group allocation was not randomized and there is no experimental design, no statement can be made about the effectiveness of the coaching intervention.

The assignment of the coaching participants to the groups “participants without mental illness” and “participants with mental illness” was based on standardized screening questions and self-reports by the participants (treatments, current practitioners). This global exploration question contains the central criteria of any mental disorder, i.e., strong and enduring suffering of mental health symptoms (anxiety, mood, interactional problems) and life and work participation problems and potentially treatment. This type of short exploration has been proven valid for the correct recognition of persons with mental disorders in earlier studies [113]. To increase external validity in future studies, the use of standardized measurement instruments for mental disorders is recommended.

The coach had behavior therapeutic expertise and was additionally supervised by a social medicine trained senior therapist to monitor the classification of all coaching participants into “participants with mental illness” and “participants without mental illness” and to supervise all coaching processes.

Since all coaching sessions were conducted by the same coach, there is a high degree of comparability. Since little is known about coaching with participants with mental illness [58,59], this study makes an important novel contribution to this research topic. It should be considered that the sample already brought a high functional level compared to other samples [93] in the pre-measure and even further improvements could be seen in both groups. In the absence of a control group, the statements about changes in work resources should not be interpreted as evidence of effectiveness.

The positive effects of the coaching were not confirmed at follow-up. This is due to the methodological fact that the sample was greatly reduced: the online survey two weeks after the coaching had a response rate of 15% (*N* = 30), so the results are based on a small sample size and should be interpreted with caution. The low response rate could be due to the fact that the pre- and post-surveys were conducted as a paper–pencil version, and T2 took place as an online survey. It is possible that the link to the survey was lost in the mass of work emails despite repeated reminders. In the future, care should be taken to ensure that follow-up measurement points are also carried out in paper–pencil form or monitored more closely to guarantee a higher response rate. It is possible that the reduction in the sample size explains the insignificance between T0 and T2.

## 5. Conclusions

In this study, a short coaching of three sessions proved useful for participants with and without mental illness in strengthening behavioral work resources (work ability, work-related capacities, coping behavior). Using situational behavioral analysis to explore one work problem and goal setting have proven useful. On this basis, coaching techniques can be selected according to the individual needs. Such individual services can be as effective or even more effective than group training [120], they should be offered at least as a supplement.

The coaching topics of participants with and without mental illness look similar at first glance but have different causes that need to be explored by the coach: are work problems the expression of (temporary or permanent) capacity impairments due to mental disorders or the reaction of a healthy employee to high work stress [72]? Coaches should be able to make this distinction through their own psychotherapeutic expertise or supervision [69]. The participants’ health status is also reflected in the unwanted events of the coaching: participants with mental illness show more negative effects in connection with the coaching (e.g., dependence on the coach), which the coach must cushion accordingly. Because coaching is not a curative treatment, subsequent referral to psychotherapy may be appropriate in justified cases [73].

Further studies should investigate the effectiveness of (short) coaching in RCT studies, considering the participants’ mental health status and focusing on behavioral characteristics such as capacities and coping. (Coping with) unwanted events should be taught in coach education to advance coaches’ professionality.

## Figures and Tables

**Figure 1 behavsci-14-00462-f001:**
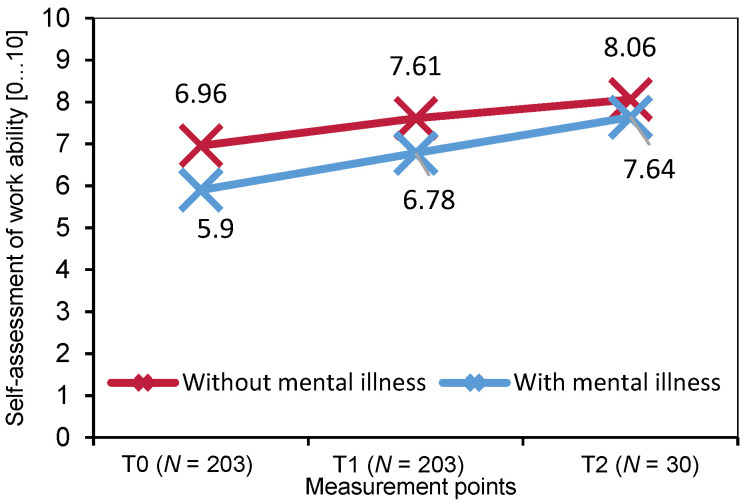
Work ability measured by Work Ability Index (WAI) pre–post coaching and two weeks after last coaching session for participants with and without mental illness (*N* = 203).

**Figure 2 behavsci-14-00462-f002:**
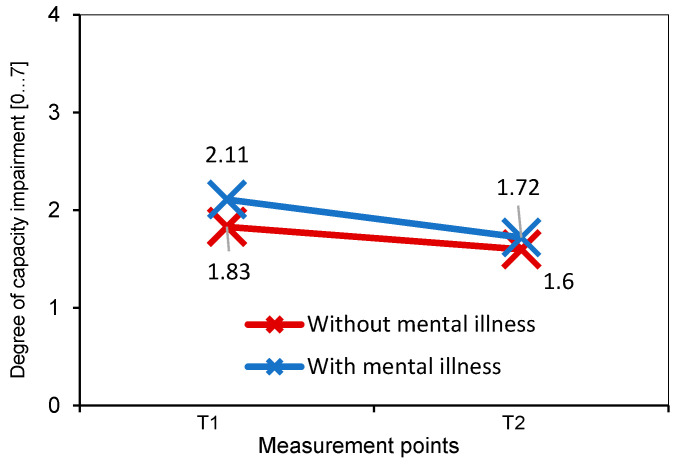
Capacity impairment measured by Mini-ICF-APP-S pre–post coaching for participants with and without mental illness (*N* = 203).

**Figure 3 behavsci-14-00462-f003:**
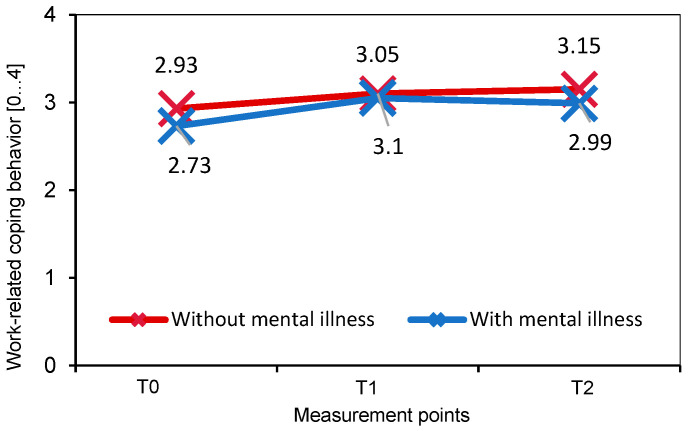
Work-related coping behavior measured by the Inventory for Job Coping (JoCoRi) during the coaching and two weeks afterwards for participants with and without mental illness (*N* = 203).

**Table 1 behavsci-14-00462-t001:** Overview of the measurement points and diagnostics.

T0		T1	T2
1. Session: Behavior Analysis	2. Session: Training	3. Session: Reflection	Two Weeks after Third Session
Work ability (WAI) Capacity impairment (Mini-ICF-APP-S) Job coping (JoCoRi)		WAI Mini-ICF-APP-S JoCoRi Side effects	WAI JoCoRi

**Table 2 behavsci-14-00462-t002:** Average age, gender distribution, professional fields, and coaching topics of participants with and without mental illness.

	Participants without Mental Illness		Participants with Mental Illness	
	*M*	*SD*	*M*	*SD*
**Average age in years**	40.41	13.05	46.06	12.00
	***n* = 103**	**%**	***n* = 100**	**%**
**Gender**				
Female	77	75	80	80
Male	26	25	20	20
**Professional fields**				
Office	33	32	29	29
Service	31	30	27	27
Education and research	26	25	20	20
Healthcare	9	9	18	18
Production	4	4	6	6
**Coaching topics**				
Role stressors	7	6.5	4	4
Workload	34	33	33	33
Situational constraints	4	4	4	4
Lack of control	5	5	3	3
Interpersonal demands	37	36	37	37
Careers issues	13	12.5	10	10
Job conditions	3	3	9	9
Acute stressors	0	0	0	0

Notes: Taxonomy of coaching topics adapted from [112]; types of professions inspired by [113]. Exact test according to Fisher: there are no significant differences in the distribution of professional fields (*p* = 0.268) and coaching topics (*p* = 0.572) of coaching participants with and without mental illness.

**Table 3 behavsci-14-00462-t003:** Goal attainment rated by participants and coach (*N* = 203).

	Participants without Mental Illness (*n* = 103)		Participants with Mental Illness (*n* = 100)	
**Goal Attainment—Participants**	** *n* **	**%**	** *n* **	**%**
Much less than expected	0	0	1	1
Less than expected	10	10	3	3
Expected result	62	60	67	67
More than expected	29	28	27	27
Much more than expected	2	2	2	2
**Goal Attainment—Coach**	** *n* **	**%**	** *n* **	**%**
Much less than expected	0	0	2	2
Less than expected	2	2	24	24
Expected result	13	13	57	57
More than expected	71	69	16	16
Much more than expected	17	16	1	1

Notes: Fisher’s exact test for differences in the frequency distribution of goal attainment: there are no significant differences in the frequency of goal attainment between participants with and without mental illness (participants *p* = 0.318; coach *p* = 0.354).

**Table 4 behavsci-14-00462-t004:** Unwanted events in coaching (*N* = 203).

	Participants without Mental Illness (*n* = 103)		Participants with Mental Illness (*n* = 100)		χ^2^-Test
**Unwanted Events**	** *n* **	**%**	** *n* **	**%**	** *p* **
No unwanted events occurred	41	40	33	33	0.781
Worsening of existing complaints	16	15.5	15	15	0.885
Occurrence of new complaints	16	15.5	16	16	0.817
Aggravation of problems	21	20	23	23	0.626
Dependence on the coach	4	4	20	20	<0.001
Discomfort during coaching sessions	7	7	10	10	0.527
Problems with coaching requirements	6	6	10	10	0.296
Problems with the coach	0	0	0	0	0.738
Problems in partnership and family	9	9	6	6	0.799
Problems with friends, neighbors, or others	3	3	2	2	0.913
Problems at work	26	25	22	22	0.901
Dissatisfaction with coaching result so far	6	6	1	1	0.211
More sessions needed	15	14.5	24	24	0.106
Other negative developments in life	31	30	14	14	0.044
Problems with third-party comments	3	3	1	1	0.668

## Data Availability

The data can be obtained on request from the authors.

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
