# Peer review of "Effects and Side Effects in a Short Work Coaching for Participants with and without Mental Illness"

_behavsci, 2024, doi:10.3390/bs14060462_

Round 1
Reviewer 1 Report
Comments and Suggestions for Authors
The authors present a study in which the effects of coaching on work-related issues in mentally healthy and mentally ill employees are examined. This study is interesting as it contributes to the body of knowledge on the benefits of coaching approaches for people diagnosed with a mental illness in the work environment. However, the manuscript has some shortcomings and I suggest minor revisions to improve its quality and make it suitable for publication in this journal.
Below are my comments on the sections of the manuscript that need improvement.
Abstract
The abstract covers all the main aspects of the study. Just one small comment:
Lines 7-8: High workload is only one of many factors that can lead to psychosocial risks, so I suggest adding “for example, high workload”.
Introduction
The introduction would benefit from a definition of work ability.
Lines 69-70: what are the differences between internal and external coaches?
Materials and Methods
It is not entirely clear how people with a mental health diagnosis were identified. Were special tests carried out to determine the diagnosis, or did the authors rely solely on the self-reporting of the participants?
It is also not clear in which language the measurement instruments were used. If the German version was used, did the authors use a version of each instrument validated in German?
The response rate at T2 is very low. Can the authors give possible reasons for this?
References
Authors should reduce the number of self-citations.
Comments on the Quality of English LanguageMinor editing of English language required.
Author Response
Dear reviewers,
Thank you very much for your detailed and helpful comments on our coaching evaluation for people with and without mental health problems! We have revised the work according to your your comments as follows (see attached document).
Best regards,
The Authors

Reviewer 2 Report
Comments and Suggestions for Authors
It is an interesting topic to compare mentally ill cohort to healthy participant with regard of efficacy and safety balance. As the authors pointed out, work-oriented short coaching are more easily accessible than psychotherapy, so it would be helpful to understand the consequence and tailor the training to obtain a better efficacy-to-safety profile.
All research questions are comparing coaching results (both efficacy and safety) between mentally ill participants with mentally healthy participants. However, the "treatment" (work-oriented short coaching) is identical between cohorts. So it becomes a prospective observational study, with two study cohorts. As an observational study, the two cohorts differ slightly according to Table 2. Usually some form of matching and adjustment (such as propensity score) will be required to generate valid comparisions.
Again, all four research questions are comparing coaching results (both efficacy and safety) between mentally ill participants with mentally healthy participants. However, in the abstract, the authors did not mention the comparison results, but rather described improvements and side effects within each cohort, which is extremely confusing.
In addition, the authors mentioned the drop-out situation at T2 is severe (N=30/203~15%). Extreme reporting bias and selection bias would raise from such missing data. I do not see effort to address this missing data. Using complete data assumes the participants left are the same as those who responded in T2. This assumption needs to be verified by some other models as sensitivity analyses. There are quite some simple methods for it, longitudinal models (mixed effect model) is one of them, which better fits the data given the repeatedly measured outcome. The insignificance between T0-T2 might simply because of low sample size, considering covariance within each participant in longitudinal models can also help with the power loss.
What's also puzzling is that the authors claimed to use "multivariate analysis of variance", but the degree of freedom reported in F statistics are only N-1. Also, F distribution by definition has 2 dimensions of degrees of freedom (df1 and df2), I do not quite understand the F(202) part.
Also, the way the authors showed "mental health status interacts with the measurement time point" is not solid. A significant difference at T0 and an insignificant difference at T1 does not guarantee significant interaction. The interaction term needs to be considered in the model, or the change from baseline (T0-T1) needs to be compared between cohorts directly.
Finally, the generalizability (external validity) is questionable. The term "mentally ill" is way to broad to characterize a population. All we know is that those participants can still work. Efforts are needed to address this limitation.
Overall, it is an interesting research however the sample selection and statistical methods are not strong enough to answer the study questions. The way this paper is reported, did not directly answer the study questions either.
Author Response

(The authors gave the same response as above.)

Round 2
Reviewer 2 Report
Comments and Suggestions for Authors
The statistical exploration and strategy is still underwhelming. Given the sample size and missing data issue, I think it makes sense to include those limitations in the manuscript and share the results on the relevant research community, as the authors did in this revision. I still think a propensity score based method would be needed, but given it's not a epidemiology centered paper, I will not hold the authors against it. Please consult a statistician or epidemiologist next time.